# JOINT AUTOENCODERS: A FLEXIBLE META-LEARNING FRAMEWORK

## ABSTRACT

The incorporation of prior knowledge into learning is essential in achieving good performance based on small noisy samples. Such knowledge is often incorporated through the availability of related data arising from domains and tasks similar to the one of current interest. Ideally one would like to allow both the data for the current task and for previous related tasks to self-organize the learning system in such a way that commonalities and differences between the tasks are learned in a data-driven fashion. We develop a framework for learning multiple tasks simultaneously, based on sharing features that are common to all tasks, achieved through the use of a modular deep feedforward neural network consisting of shared branches, dealing with the common features of all tasks, and private branches, learning the specific unique aspects of each task. Once an appropriate weight sharing architecture has been established, learning takes place through standard algorithms for feedforward networks, e.g., stochastic gradient descent and its variations. The method deals with meta-learning (such as domain adaptation, transfer and multi-task learning) in a unified fashion, and can easily deal with data arising from different types of sources. Numerical experiments demonstrate the effectiveness of learning in domain adaptation and transfer learning setups, and provide evidence for the flexible and task-oriented representations arising in the network.

## 1 INTRODUCTION

A major goal of inductive learning is the selection of a rule that generalizes well based on a finite set of examples. It is well-known ((Hume, 1748)) that inductive learning is impossible unless some regularity assumptions are made about the world. Such assumptions, by their nature, go beyond the data, and are based on prior knowledge achieved through previous interactions with 'similar' problems. Following its early origins ((Baxter, 2000; Thrun and Pratt, 1998)), the incorporation of prior knowledge into learning has become a major effort recently, and is gaining increasing success by relying on the rich representational flexibility available through current deep learning schemes (Bengio et al., 2013). Various aspects of prior knowledge are captured in different settings of meta-learning, such as learning-to-learn, domain adaptation, transfer learning, multi-task learning, etc. (e.g., (Goodfellow et al., 2016)). In this work, we consider the setup of multi-task learning, first formalized in (Baxter, 2000), where a set of tasks is available for learning, and the objective is to extract knowledge from a subset of tasks in order to facilitate learning of other, related, tasks. Within the framework of representation learning, the core idea is that of shared representations, allowing a given task to benefit from what has been learned from other tasks, since the shared aspects of the representation are based on more information (Zhang et al., 2008).

We consider both unsupervised and semi-supervised learning setups. In the former setting we have several related datasets, arising from possibly different domains, and aim to compress each dataset based on features that are shared between the datasets, and on features that are unique to each problem. Neither the shared nor the individual features are given apriori, but are learned using a deep neural network architecture within an autoencoding scheme. While such a joint representation could, in principle, serve as a basis for supervised learning, it has become increasingly evident that representations should contain some information about the output (label) identity in order to perform well, and that using pre-training based on unlabeled data is not always advantageous (e.g., chap. 15 in (Goodfellow et al., 2016)). However, since unlabeled data is far more abundant than labeled data, much useful information can be gained from it. We therefore propose a joint encoding-classification

scheme where both labeled and unlabeled data are used for the multiple tasks, so that internal representations found reflect both types of data, but are learned simultaneously.

**The main contributions of this work are:** *(i)* A generic and flexible modular setup for combining unsupervised, supervised and transfer learning. *(ii)* Efficient end-to-end transfer learning using mostly unsupervised data (i.e., very few labeled examples are required for successful transfer learning). *(iii)* Explicit extraction of task-specific and shared representations.

## 2    RELATED WORK

Previous related work can be broadly separated into two classes of models: *(i)* Generative models attempting to learn the input representations. *(ii)* Non-generative methods that construct separate or shared representations in a bottom-up fashion driven by the inputs.

We first discuss several works within the non-generative setting. The Deep Domain Confusion (DDC) algorithm in (Tzeng et al., 2014) studies the problems of unsupervised domain adaptation based on sets of unlabeled samples from the source and target domains, and supervised domain adaptation where a (usually small) subset of the target domain is labeled . By incorporating an adaptation layer and a domain confusion loss they learn a representation that optimizes both classification accuracy and domain invariance, where the latter is achieved by minimizing an appropriate discrepancy measure. By maintaining a small distance between the source and target representations, the classifier makes good use of the relevant prior knowledge. The algorithm suggested in (Ganin and Lempitsky, 2015) augments standard deep learning with a domain classifier that is connected to the feature extractor, and acts to modify the gradient during backpropagation. This adaptation promotes the similarity between the feature distributions in a domain adaptation task. The Deep Reconstruction Classification Network (DRCN) in (Ghifary et al., 2016) tackles the unsupervised domain adaptation task by jointly learning a shared encoding representation of the source and target domains based on minimizing a loss function that balances between the classification loss of the (labeled) source data and the reconstruction cost of the target data. The shared encoding parameters allow the target representation to benefit from the ample source supervised data. In addition to these mostly algorithmic approaches, a number of theoretical papers have attempted to provide a deeper understanding of the benefits available within this setting (Ben-David et al., 2009; Maurer et al., 2016).

Next, we mention some recent work within the generative approach, briefly. Recent work has suggested several extensions of the increasingly popular Generative Adversarial Networks (GAN) framework (Goodfellow et al., 2014). The Coupled Generative Adversarial Network (CoGAN) framework in (Liu and Tuzel, 2016) aims to generate pairs of corresponding representations from inputs arising from different domains. They propose learning joint distributions over two domains based only on samples from the marginals. This yields good results for small datasets, but is unfortunately challenging to achieve for large adaptation tasks, and is computationally cumbersome. The Adversarial Discriminative Domain Adaptation (ADDA) approach (Tzeng et al., 2017) subsumes some previous results within the GAN framework of domain adaptation. The approach learns a discriminative representation using the data in the labeled source domain, and then learns to adapt the model for use in the (unlabeled) target domain through a domain adversarial loss function. The idea is implemented through a minimax formulation similar to the original GAN setup.

The extraction of shared and task-specific representations is the subject of a number of works, such as (Evgeniou and Pontil, 2004) and (Parameswaran and Weinberger, 2010). However, works in this direction typically require inputs of the same dimension and for the sizes of their shared and task-specific features to be the same.

A great deal of work has been devoted to multi-modal learning where the inputs arise from different modalities. Exploiting data from multiple sources (or views) to extract meaningful features, is often done by seeking representations that are sensitive only to the common variability in the views and are indifferent to view-specific variations. Many methods in this category attempt to maximize the correlation between the learned representations, as in the linear canonical correlation analysis (CCA) technique and its various nonlinear extensions (Andrew et al., 2013; Michaeli et al., 2016). Other methods use losses based on both correlation and reconstruction error (in an auto-encoding like scheme) (Wang et al., 2015), or employ diffusion processes to reveal the common underlying

manifold (Lederman and Talmon, 2015). However, all multi-view representation learning algorithms rely on *paired examples* from the two views. This setting is thus very different from transfer learning, multi-task learning, or domain adaptation, where one has access only to *unpaired samples* from each of the domains.

While GANs provide a powerful approach to multi-task learning and domain adaptation, they are often hard to train and fine tune ((Goodfellow, 2016)). Our approach offers a complementary non-generative perspective, and operates in an end-to-end fashion allowing the parallel training of multiple tasks, incorporating both unsupervised, supervised and transfer settings within a single architecture. This simplicity allows the utilization of standard optimization techniques for regular deep feedforward networks, so that any advances in that domain translate directly into improvements in our results. The approach does not require paired inputs and can operate with inputs arising from entirely different domains, such as speech and audio (although this has not been demonstrated empirically here). Our work is closest to (Bousmalis et al., 2016)which shares with us the separation into common and private branches. They base their optimization on several loss functions beyond the reconstruction and classification losses, enforcing constraints on intermediate representations. Specifically, they penalize differences between the common and private branches of the same task, and encourage similarity between the different representations of the source and target in the common branch. This multiplicity of loss functions adds several free parameters to the problem that require further fine-tuning. Our framework uses only losses penalizing reconstruction and classification errors, thereby directly focusing on the task without adding internal constrains. Moreover, since DSN does not use a classification error for the target it cannot use labeled targets, and thus can only perform unsupervised transfer learning. Also, due to the internal loss functions, it is not clear how to extend DSN to multi-task learning, which is immediate in our formalism. Practically, the proposed DSN architecture is costly; it is larger by more than on order of magnitude than either the models we have studied or ADDA. Thus it is computationally challenging as well as relatively struggling to deal with small datasets.

## 3 JOINT AUTOENCODERS

In this section, we introduce *joint autoencoders* (JAE), a general method for multi-task learning by unsupervised extraction of features shared by the tasks as well as features specific to each task. We begin by presenting a simple case, point out the various possible generalizations, and finally describe two transfer and multi-task learning procedures utilizing joint autoencoders.

### 3.1 JOINT AUTOENCODERS FOR RECONSTRUCTION

Consider a multi-task learning scenario with $T$ tasks $t^1, ..., t^T$ defined by domains $\left\{ \left( \mathcal{X}^i \right) \right\}_{i=1}^{T}$. Each task $t^i$ is equipped with a set of unlabeled samples $\left\{ x_n^i \in \mathcal{X}^i \right\}_{n=1}^{N^{i,u}}$ ,where $N^{i,u}$ denotes the size of the unlabeled data set, and with a reconstruction loss function $\ell_r^i \left( x_n^i, \tilde{x}_n^i \right)$, where $\tilde{x}_n^i$ is the reconstruction of the sample $x_n^i$. Throughout the paper, we will interpret $\ell_r^i$ as the $L_2$ distance between $x_n^i$ and $\tilde{x}_n^i$, but in principle $\ell_r^i$ can represent any unsupervised learning goal. The tasks are assumed to be related, and we are interested in exploiting this similarity to improve the reconstruction. To do this, we make the following two observations:

*(i)* Certain aspects of the unsupervised tasks we are facing may be similar, but other aspects may be quite different (e.g., when two domains contain color and grayscale images, respectively).

*(ii)* The similarity between the tasks can be rather "deep". For example, cartoon images and natural images may benefit from different low-level features, but may certainly share high-level structures. To accommodate these two observations, we associate with each task $t^i$ a pair of functions: $f_p^i \left( x; \theta_p^i \right)$, the "private branch", and $f_s^i \left( x; \theta_s^i, \tilde{\theta}_s \right)$, the "shared branch" . The functions $f_p^i$ are responsible for the task-specific representations of $t^i$ and are parametrized by parameters $\theta_p^i$. The functions $f_s^i$ are responsible for the shared representations, and are parametrized, in addition to parameters $\theta_s^i$, by $\tilde{\theta}_s$ shared by all tasks. The key idea is that the weight sharing forces the common branches to learn to represent the common features of the two sources. Consequently, the private branches are implicitly forced to capture only the features that are not common to the other task. We aim at minimizing the

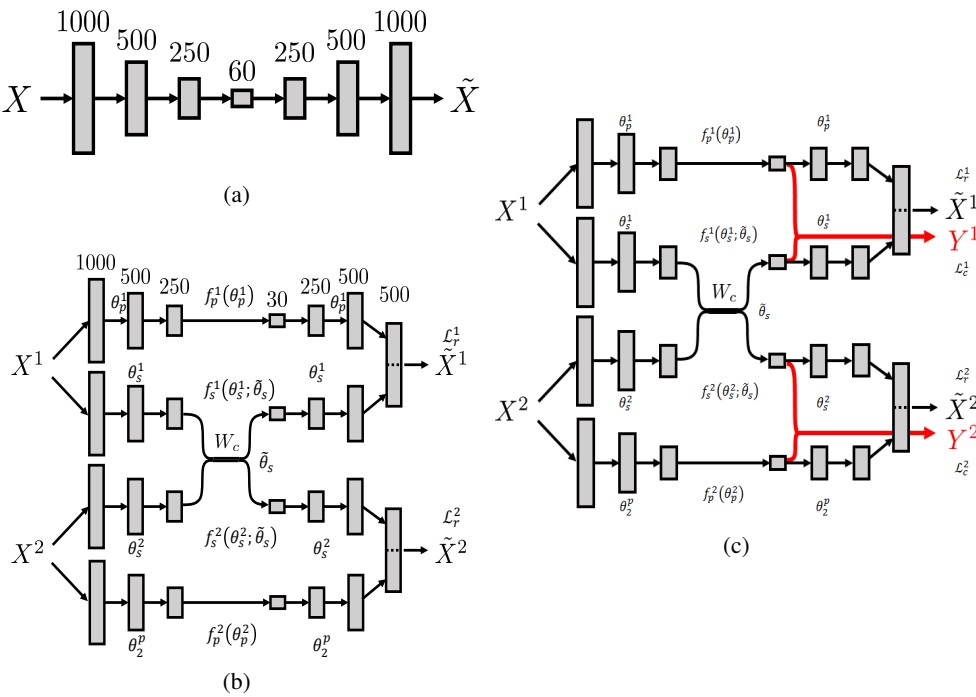

Figure 1: (a) An example of an MNIST autoencoder (b) The joint autoencoder constructed out of the AE in (a), where $X^1 = \{0, 1, 2, 3, 4\}$ and $X^2 = \{5, 6, 7, 8, 9\}$. Each layer is a fully connected one, of the specified size, with ReLU activations. The weights shared by the two parts are denoted by $W_c$. The pairs of the top fully connected layers of dimension 500 are concatenated to create a layer of dimension 1000 which is then used directly to reconstruct the input of size 784. (c) A schematic depiction of a JAE architecture extended for supervised learning. The parameters and functions in figures (b) and (c) are explained in the main text.

cumulative weighted loss

$$\mathcal{L}_r = \sum_{i=1}^{T} w_r^i \sum_{n=1}^{N^{i,u}} \ell_r^i \left( x_n^i, f_p^i \left( x_n^i; \theta_p^i \right), f_s^i \left( x_n^i; \theta_s^i, \tilde{\theta}_s \right) \right). \tag{1}$$

In practice, we implement all functions as autoencoders and the shared parameters $\tilde{\theta}_s$ as the bottleneck of the shared branch of each task, with identical weights across the tasks. Our framework, however, supports more flexible sharing as well, such as sharing more than a single layer, or even partially shared layers. The resulting network can be trained with standard backpropagation on all reconstruction losses simultaneously. Figure 1(a) illustrates a typical autoencoder for the MNIST dataset, and Figure 1(b) illustrates the architecture obtained from implementing all branches in the formal description above with such autoencoders (AE). We call this architecture a *joint autoencoder* (JAE).

As mentioned before, in this simple example, both inputs are MNIST digits, all branches have the same architecture, and the bottlenecks are single layers of the same dimension. However, this need not be the case. The inputs can be entirely different (e.g., image and text), all branches may have different architectures, the bottleneck sizes can vary, and more than a single layer can be shared. Furthermore, the shared layers need not be the bottlenecks, in general. Finally, the generalization to more than two tasks is straightforward - we simply add a pair of autoencoders for each task, and share some of the layers of the common-feature autoencoders. Weight sharing can take place between subsets of tasks, and can occur at different levels for the different tasks.

## 3.2 Joint Autoencoders for Multi-Task, Semi-Supervised and Transfer Learning

Consider now a situation in which, in addition to the unlabeled samples from all domains $\mathcal{X}^i$, we also have datasets of labeled pairs $\left\{ \left( x_k^i, y_k^i \right) \right\}_{k=1}^{N^{i,l}}$ where $N^{i,l}$ is the size of the labeled set for task $t^i$ and is assumed to be much smaller than $N^{i,u}$. The supervised component of each task $t^i$ is reflected in the supervised loss $\ell_c^i \left( y_n^i, \tilde{y}_n^i \right)$, typically multi-class classification. We extend our loss definition in Equation 1 to be

$$\mathcal{L} = \mathcal{L}_r + \mathcal{L}_c = \mathcal{L}_r + \sum_{i=1}^{T} w_c^i \sum_{n=1}^{N^{i,l}} \ell_c^i \left( y_n^i, f_p^i \left( x_n^i; \theta_p^i \right), f_s^i \left( x_n^i; \theta_s^i, \tilde{\theta}_s \right) \right), \tag{2}$$

where we now interpret the functions $f_s^i, f_p^i$ to also output a classification. Figure 1(c) illustrates the schematic structure of a JAE extended to include supervised losses. Note that this framework supports various learning scenarios. Indeed, if a subset of the tasks has $N^{i,l} = 0$, the problem becomes one of unsupervised domain adaptation. The case where $N^{i,l}$ are all or mostly small describes semi-supervised learning. If some of the labeled sets are large while the others are either small or empty, we find ourselves facing a transfer learning challenge. Finally, when all labeled sets are of comparable sizes, this is multi-task learning, either supervised (when $N^{i,l}$ are all positive) or unsupervised (when $N^{i,l} = 0$).

We describe two strategies to improve supervised learning by exploiting shared features.

**Common-branch transfer**   In this approach, we first train joint autoencoders on both source and target tasks simultaneously, using all available unlabeled data. Then, for the source tasks (the ones with more labeled examples), we fine-tune the branches up to the shared layer using the sets of labeled samples, and freeze the learned shared layers. Finally, for the target tasks, we use the available labeled data to train only its private branches while fixing the shared layers fine-tuned on the source data.

**End-to-end learning**   The second, *end-to-end* approach, combines supervised and unsupervised training. Here we extend the JAE architecture by adding new layers, with supervised loss functions for each task; see Figure 1(c). We train the new network using all losses from all tasks simultaneously - reconstruction losses using unlabeled data, and supervised losses using labeled data. When the size of the labeled sets is highly non-uniform, the network is naturally suitable for transfer learning. When the labeled sample sizes are roughly of the same order of magnitude, the setup is suitable for semi-supervised learning.

## 3.3 On the Depth of Sharing

It is common knowledge that similar *low-level features* are often helpful for similar tasks. For example, in many vision applications, CNNs exhibit the same Gabor-type filters in their first layer, regardless of the objects they are trained to classify. This observation makes low-level features immediate candidates for sharing in multi-task learning settings. However, unsurprisingly, sharing low-level features is not as beneficial when working with domains of different nature (e.g., handwritten digits vs. street signs). Our approach allows to share weights in deeper layers of a neural net, while leaving the shallow layers un-linked. The key idea is that by forcing all shared-branch nets to share deep weights, their preceding shallow layers must learn to transform the data from the different domains into a common form. We support this intuition through several experiments. As our preliminary results in Section 4.2.1 show, for similar domains, sharing deep layers provides the same performance boost as sharing shallow layers. Thus, we pay no price for relying only on "deep similarities". But for domains of a different nature, sharing deep layers has a clear advantage.

## 4 Experiments

All experiments were implemented in Keras over Tensorflow. The code will be made available soon, and the network architectures used are given in detail in the appendix.

### 4.1 UNSUPERVISED LEARNING

We present experimental results demonstrating the improvement in unsupervised learning of multiple tasks on the MNIST and CIFAR-10 datasets. For the MNIST experiment, we have separated the training images into two subsets: $X^1$, containing the digits $\{0-4\}$ and $X^2$, containing the digits $\{5-9\}$. We compared the $L_2$ reconstruction error achieved by the JAE to a baseline of a pair of AEs trained on each dataset with architecture identical to a single branch of the JAE. The joint autoencoder (MSE =5.4) out-performed the baseline (MSE = 5.6) by $4\%$. The autoencoders had the same cumulative bottleneck size as the JAE, to ensure the same hidden representation size. To ensure we did not benefit solely from increased capacity, we also compared the AEs to a JAE with the same total number of parameters as the baseline, obtained by reducing the size of each layer by $\sqrt{2}$. This model achieved an MSE of $5.52$, a $1.4\%$ improvement over the baseline.

To further understand the features learned by the shared and private bottlenecks, we visualize the activations of the bottlenecks on $1000$ samples from each dataset, using $2D$ t-SNE embeddings (van der Maaten and Hinton, 2008). Figure 2(a) demonstrates that the common branches containing the shared layer (green and magenta) are much more mixed between themselves than the private branches (red and black), indicating that they indeed extract shared features. Figure 2(b) displays examples of digits reconstructions. The columns show (from left to right) the original digit, the image reconstructed by the full JAE, the output of the private branches and the shared branches. We see that the common branches capture the general shape of the digit, while the private branches capture the fine details which are specific to each subset.

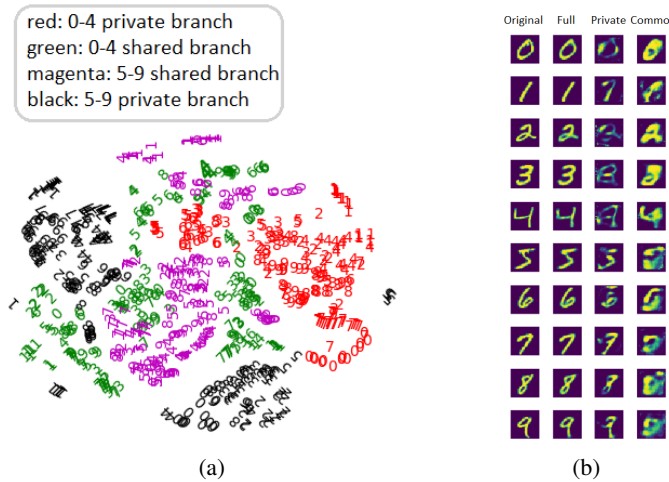

(a)            (b)

Figure 2: (a) t-SNE visualizations of the responses of each bottleneck to images from $\{0-4\}$ and $\{5-9\}$ MNIST digits: red and black for the private branches of the datasets, green and magenta for the shared branches. The digits from different branches in the figure are rotated to avoid clutter and occlusion (b) From left to right: original digits, reconstruction by the JAE, reconstruction by the private branch, reconstruction by the shared branch.

We verify quantitatively the claim about the differences in separation between the private and shared branches. The Fisher criterion for the separation between the t-SNE embeddings of the private branches is $7.22 \cdot 10^{-4}$, whereas its counterpart for the shared branches is $2.77 \cdot 10^{-4}$, 2.6 times less. Moreover, the shared branch embedding variance for both datasets is approximately identical, whereas the private branches map the dataset they were trained on to locations with variance greater by $1.35$ than the dataset they had no access to. This illustrates the extent to which the shared branches learn to separate both datasets better than the private ones.

For CIFAR-10 we trained the baseline autoencoder on single-class subsets of the database (e.g., all airplane images) and trained the JAE on pairs of such subsets. Table 1 shows a few typical results, demonstrating a consistent advantage for JAEs. Besides the lower reconstruction error, we can see

that visually similar image classes, enjoy a greater boost in performance. For instance, the pair deer-horses enjoyed a performance boost of $37\%$, greater than the typical boost of $33 - 35\%$. As with MNIST, we also compared the pair of autoencoders to a JAE with the same total number of parameters (obtained by $\sqrt{2}$ size reduction of each layer), achieving $22 - 24\%$ boost. Thus, the observed improvement is clearly not a result of mere increased network capacity.

Table 1: JAE reconstruction performance

|                         | A-D  | A-H  | A-S  | D-H  | D-S  | H-S  |
|-------------------------|------|------|------|------|------|------|
| AE error                | 20.8 | 18.5 | 16.2 | 20.6 | 18.2 | 16.0 |
| JAE error               | 13.9 | 12.2 | 10.8 | 13.2 | 11.4 | 10.6 |
| JAE-reduced error       | 16.2 | 14.2 | 12.6 | 15.6 | 14.0 | 12.3 |
| JAE Improvement         | 33%  | 34%  | 33%  | 37%  | 35%  | 34%  |
| JAE-reduced Improvement | 22%  | 23%  | 22%  | 24%  | 23%  | 23%  |

Performance of JAEs and JAEs reduced by a $\sqrt{2}$ factor vs standard AEs in terms of reconstruction MSE on pairs of objects in CIFAR-10: airplanes (A), deer (D), horses (H), ships(S). For each pair of objects, we give the standard AE error, JAE and JAE-reduced error and the improvement percentage.

We remark that we experimented with an extension of unsupervised JAEs to variational autoencoders ((Kingma and Welling, 2014)). Unlike standard VAEs, we trained three hidden code layers, requiring each to have a small Kullback-Leibler divergence from a given normal distribution. One of these layers was used to reconstruct both datasets (analogous to the shared bottleneck in a JAE), while the other two were dedicated each to one of the datasets (analogous to the private branches). The reconstruction results on the halves of the MNIST dataset were promising, yielding an improvement of $12\%$ over a pair of VAEs of the same cumulative size. Unfortunately, we were not able to achieve similar results on the CIFAR-10 dataset, nor to perform efficient multi-task\ transfer learning with joint VAEs. This remains an intriguing project for the future,

## 4.2 TRANSFER LEARNING

Next, we compare the performance on MNIST of the two JAE-based transfer learning methods detailed in Section 3.2. For both methods, $X^1$ contains digits from $\{0 - 4\}$ and $X^2$ contains the digits $\{5 - 9\}$. The source and target datasets comprise $2000$ and $500$ samples, respectively. All results are measured on the full MNIST test set. The common-branch transfer method yields $92.3\%$ and $96.1\%$ classification precision for the $X^1 \rightarrow X^2$ and $X^2 \rightarrow X^1$ transfer tasks, respectively. The end-to-end approach results in $96.6\%$ and $98.3\%$ scores on the same tasks, which demonstrates the superiority of the end-to-end approach.

### 4.2.1 SHARED LAYER DEPTH

We investigate the influence of shared layer depth on the transfer performance. We see in Table 2 that for highly similar pairs of tasks such as the two halves of the MNIST dataset, the depth has little significance, while for dissimilar pairs such as MNIST-USPS, "deeper is better" - the performance improves with the shared layer depth. Moreover, when the input dimensions differ, early sharing is impossible - the data must first be transformed to have the same dimensions.

### 4.2.2 MNIST, USPS AND SVHN DIGITS DATASETS

We have seen that the end-to-end JAE-with-transfer algorithm outperforms the alternative approach. We now compare it to other domain adaptation methods that use little to no target samples for

Table 2: Shared Layer Depth and Transferability

|                                    | 1    | 2    | 3    | 4    | 5    |
|------------------------------------|------|------|------|------|------|
| MNIST {0-4} $\rightarrow$ {5-9}    | 96.5 | 95.4 | 95.8 | 96.1 | 96.0 |
| MNIST {5-9} $\rightarrow$ {0-4}    | 98.3 | 97.6 | 97.8 | 98.2 | 98.3 |
| MNIST $\rightarrow$ USPS           |      |      | 84.8 |      | 87.6 |
| USPS $\rightarrow$ MNIST           |      |      | 83.2 |      | 86.9 |

Influence of the shared layer depth on the transfer learning performance. For the MNIST-USPS pair, only partial data are available for dimensional reasons.

supervised learning, applied to the MNIST, USPS and SVHN digits datasets. The transfer tasks we consider are MNIST→USPS , USPS→MNIST and SVHN→MNIST. Following (Tzeng et al., 2017) and (Long et al., 2013), we use $2000$ samples for MNIST and $1800$ samples from USPS. For SVHN→MNIST, we use the complete training sets. In all three tasks, both the source and the target samples are used for the unsupervised JAE training. In addition, the source samples are used for the source supervised element of the network. We study the weakly-supervised performance of JAE and ADDA allowing access to a small number of target samples, ranging from $5$ to $50$ per digit. For the supervised version of ADDA, we fine-tune the classifiers using the small labeled target sets after the domain adaptation. Figure 3 $(a) - (c)$ provides the results of our experiments. For recent methods such as CoGAN, gradient reversal, domain confusion and DSN, we display results with zero supervision, as they do not support weakly-supervised training. For DSN, we provide preliminary results on MNIST↔USPS, without model optimization that is likely to prevent over-fitting.

On all tasks, we achieve results comparable or superior to existing methods using very limited supervision, despite JAE being both conceptually and computationally simpler than competing approaches. In particular, we do not train a GAN as in CoGAN, and require a single end-to-end training period, unlike ADDA that trains three separate networks in three steps. Computationally, the models used for MNIST→USPS and USPS→MNIST have $1.36M$ parameters, whereas ADDA uses over $1.5M$ weights. For SVHN→MNIST, we use a model with $3M$ weights, comparable to the $1.5M$ parameters in ADDA and smaller by an order of magnitude than DSN. The SVHN→MNIST task is considered the hardest (for instance, GAN-based approaches fail to address it) yet the abundance of unsupervised training data allows us to achieve good results, relative to previous methods. We provide further demonstration that knowledge is indeed transferred from the source to the target in the MNIST→USPS transfer task with $50$ samples per digit. Source supervised learning, target unsupervised learning and target classifier training are frozen after the source classifier saturates (epoch $4$). The subsequent target test improvement by $2\%$ is due solely to the source dataset reconstruction training, passed to the target via the shared bottleneck layer (Figure 3(d)).

### 4.2.3 THREE-WAY TRANSFER LEARNING

We demonstrate the ability to extend our approach to multiple tasks with ease by transferring knowledge from SVHN to MNIST and USPS simultaneously. That is, we train a triple-task JAE reconstructing all three datasets, with additional supervised training on SVHN and weakly-supervised training on the target sets. All labeled samples are used for the source, while the targets use $50$ samples per digit. The results illustrate the benefits of multi-task learning: $94.5\%$ classification accuracy for MNIST, a $0.8\%$ improvement over the SVHN→MNIST task, and $88.9\%$ accuracy in UPS, a $1.2\%$ improvement over SVHN→USPS. This is consistent with unsupervised learning being useful for the classification. USPS is much smaller, thus it has a lower score, but it benefits relatively more from the presence of the other, larger, task. We stress that the extension to multiple tasks was straightforward, and indeed we did not tweak the various ' models, opting instead for previously used JAEs, with a single shared bottleneck. Most state-of-the-art transfer methods do not allow for an obvious, immediate adaptation for transfer learning between multiple tasks.

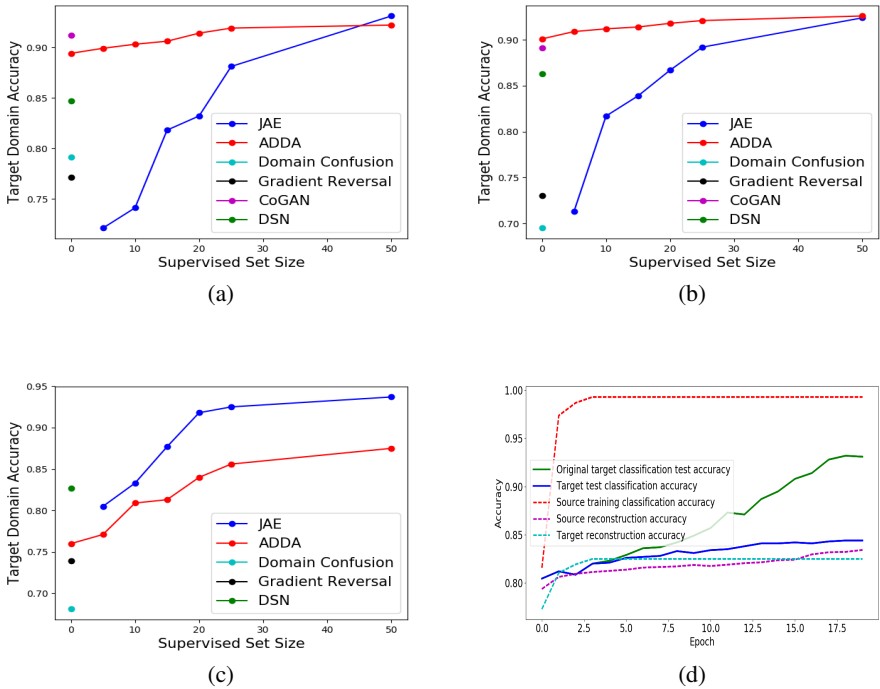

Figure 3: Transfer learning results on MNIST, USPS and SVHN. (a) MNIST→USPS. 50 samples per digit allow JAE to surpass ADDA and CoGAN, reaching $93.1\%$ accuracy. (b) USPS→MNIST. 25 samples per digit allow JAE to surpass CoGAN. At 50 samples per digit, JAE ($92.4\%$) is comparable to ADDA ($92.6\%$). (c) SVHN→MNIST. JAE achieves state-of-the-art performance, reaching 93.7% accuracy. CoGAN did not converge on this task. (d) Transfer occurring purely due to source unsupervised learning. The green graph is the test classification accuracy achieved without our interference by freezing the source supervised learning, target unsupervised learning and target classifier training. The reconstruction accuracy is measured as the fraction of pixels correctly classified as white\black. Equivalently, it is the complement of the average reconstruction error.

## 5 CONCLUSION

We presented a general scheme for incorporating prior knowledge within deep feedforward neural networks for domain adaptation, multi-task and transfer learning problems. The approach is general and flexible, operates in an end-to-end setting, and enables the system to self-organize to solve tasks based on prior or concomitant exposure to similar tasks, requiring standard gradient based optimization for learning. The basic idea of the approach is the sharing of representations for aspects which are common to all domains/tasks while maintaining private branches for task-specific features. The method is applicable to data from multiple sources and types, and has the advantage of being able to share weights at arbitrary network levels, enabling abstract levels of sharing.

We demonstrated the efficacy of our approach on several domain adaptation and transfer learning problems, and provided intuition about the meaning of the representations in various branches. In a broader context, it is well known that the imposition of structural constraints on neural networks, usually based on prior domain knowledge, can significantly enhance their performance. The prime example of this is, of course, the convolutional neural network. Our work can be viewed within that general philosophy, showing that improved functionality can be attained by the modular prior structures imposed on the system, while maintaining simple learning rules.

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

# Appendix

## A. IMPLEMENTATION DETAILS

All the images are scaled to $[0, 1]$. In all cases, the training is done using the ADAM optimizer with learning rate $10^{-3}$, $\beta_1 = 0.9$, $\beta_2 = 0.999$. The Keras default Xavier initialization is used. Shared layers are denoted in red and connected by a bidirectional arrow: $\leftrightarrow$. $Conv\, n \times (k \times k)$ stands for a convolution layer with $n$ filters of size $k \times k$. $ReLU$ stands for a rectified linear unit, i.e. the function $\max(x, 0)$. $MP\, k \times k\, stride\, l$ stands for max-pooling of size $k \times k$ with stride $l$. $FC\, k$ stands for a fully-connected layer of size $k$. The symbol $\oplus$ stands for the merge operation. For instance, if it appears after fully-connected layers of size 500 each, it denoted the resulting merged layer of size 1000. Outputs are processed by a SoftMax.

**Unsupervised learning - MNIST**

For the MNIST reconstruction experiments, we utilize a CNN-based version of the autoencoder and JAE presented in Figure 1. Mini-batch size is set to 256, with 10 epochs. The JAE losses are weighed equally.

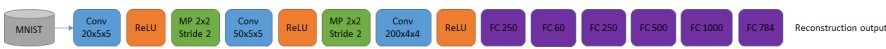

Figure 4: An MNIST autoencoder. A pair of these is used as a benchmark for the MNIST joint autoencoder.

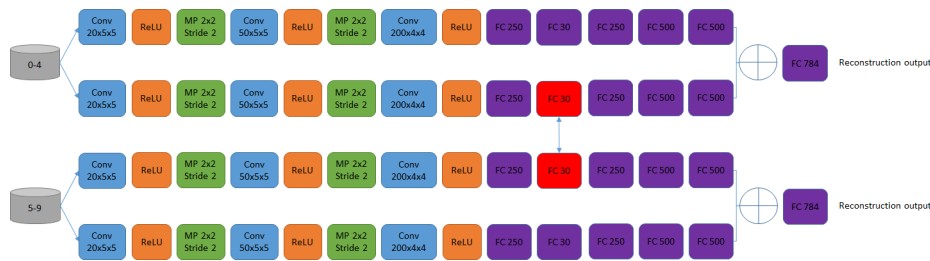

Figure 5: A joint autoencoder for reconstruction of MNIST subsets

**Unsupervised learning - CIFAR-10**

Mini-batch size is set to 128, with 10 epochs. The JAE losses are weighed equally. "Deconv $n \times k \times k$" stands for a deconvolution layer with $n$ filters of size $k \times k$ with $2 \times 2$ upsampling.

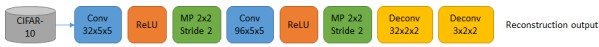

Figure 6: A CIFAR-10 autoencoder. A pair of these is used as a benchmark for the CIFAR-10 joint autoencoder.

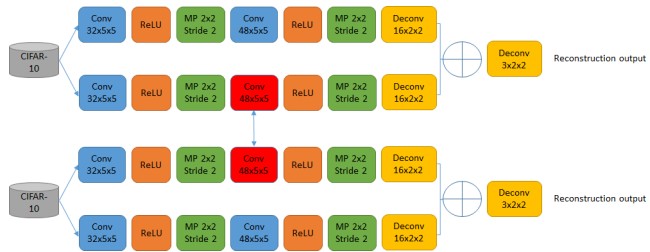

Figure 7: A joint autoencoder for reconstruction of CIFAR-10 subsets

## Transfer learning - MNIST↔USPS

Mini-batch size is set to 64, with 10 epochs. The reconstruction losses are weighed 4 times higher than the classification losses.

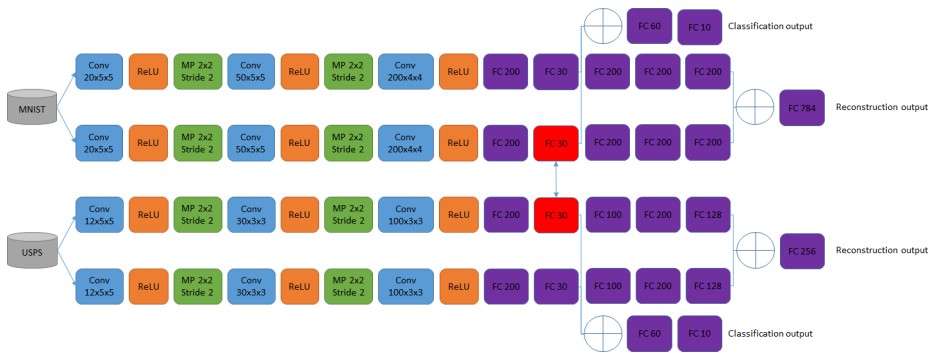

Figure 8: An MNIST-USPS joint autoencoder

## Transfer learning - SVHN→MNIST

Mini-batch size is set to 64, with 10 epochs. The reconstruction losses are weighed 4 times lower than the classification losses. In this case, as opposed to the previous one, the classification task is challenging enough to avoid early overfitting.

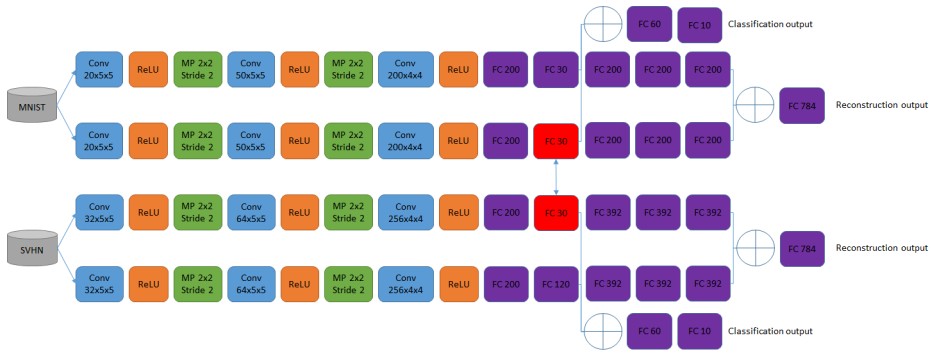

Figure 9: A joint autoencoder for transfer learning from SVHN to MNIST

## Transfer learning - SVHN→MNIST+USPS

Mini-batch size is set to $64$, with $20$ epochs. The reconstruction losses are weighed $4$ times lower than the classification losses.

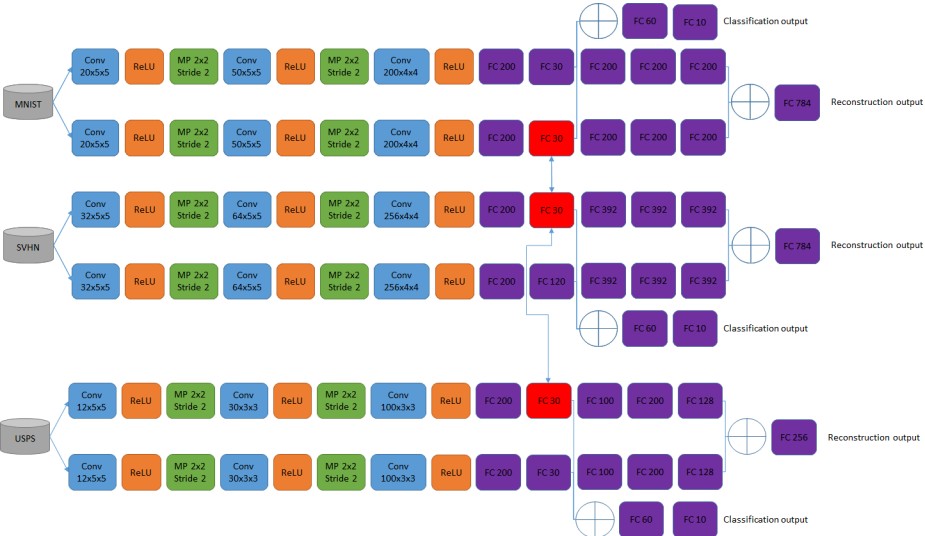

Figure 10: A three-way joint autoencoder for transfer learning from SVHN to MNIST and to USPS

