# OpenReview forum: "Joint autoencoders: a flexible meta-learning framework"
_ICLR.cc/2018/Conference — Reject_

### Official Review · AnonReviewer1 · 2017-11-28

**Rating:** 4
**Confidence:** 4

**Review:**

The work proposed a generic framework for end-to-end transfer learning / domain adaptation with deep neural networks. The idea is to learn a joint autoencoders, containing private branch with task/domain-specific weights, as well as common branch consisting of shared weights used across tasks/domains, as well as task/domain-specific weights.  Supervised losses are added after the encoders to utilize labeled samples from different tasks. Experiments on the MNIST and CIFAR datasets showed improvements over baseline models. Its performance is comparable to / worse than several existing deep domain adaptation works on the MNIST, USPS and SVHN digit datasets.

The structure of the paper is good, and easy to read.  The idea is fairly straight-forward. It reads as an extension of "frustratingly easy domain adaptation" to DNN (please cite this work). Different from most existing work on DNN for multi-task/transfer learning, which focuses on weight sharing in bottom layers, the work emphasizes the importance of weight sharing in deeper layers. The overall novelty of the work is limited though.

The authors brought up two strategies on learning the shared and private weights at the end of section 3.2. However, no follow-up comparison between the two are provided. It seems like most of the results are coming from the end-to-end learning.

Experimental results:
section 4.1: Figure 2 is flawed. The colors do not correspond to the sub-tasks. For example, there are digits 1, 4 in color magenta, which is supposed to be the shared branch of digits of 5~9. Vice versa.
From reducing the capacity of JAE to be the same as the baseline, most of the improvement is gone. It is not clear how much of the improvement will remain if the baseline model gets to see all the samples instead of just those from each sub-task.

section 4.2.1: The authors demonstrate the influence of shared layer depth in table 2. While it does seem to matter for tasks of dissimilar inputs, have the authors compare having a completely shared branch or sharing more than just a single layer?

The authors suggested in section 4.1 CIFAR experiment that the proposed method provides more performance boost when the two tasks are more similar, which seems to be contradicting to the results shown in Figure 3, where its performance is worse when transferring between USPS and MNIST, which are more similar tasks vs between SVHN and MNIST. Do the authors have any insight?

---

> ### Author Response · Authors · 2017-12-24
> **Response to AnonReviewer1**
>
> * It reads as an extension of "frustratingly easy domain adaptation" to DNN (please cite this work).
>
> There is a sizable list of references we intend to include in the next version, including the work you refer to, as well as (among others) [4], [5], [6], [7]. We attempted to comply by the "strong recommendation" to keep the references to a single page, and had to retain only those works we were explicitly influenced by, or recent state-of-the-art deep learning papers focusing on domain adaptation and explicit extraction of separate shared and task-related features.
>
> * The authors brought up two strategies on learning the shared and private weights at the end of section 3.2. However, no follow-up comparison between the two are provided. It seems like most of the results are coming from the end-to-end learning.
>
> The first paragraph in Section 4.2 provides precisely the sought-for comparison. We find that the end-to-end learning approach is both simpler and better, and thus use it for the rest of the experiments. We will add a reference to that conclusion in Section 2.3.
>
> Experiment results:
>
> * section 4.1: Figure 2 is flawed. The colors do not correspond to the sub-tasks. For example, there are digits 1, 4 in color magenta, which is supposed to be the shared branch of digits of 5~9. Vice versa.
>
> In Figure 2a, all branches are applied to all digits, with the colors representing the data that the branch was exposed to. The idea is that a branch should be more ‘inclined’ to treat digits it never saw as noise. This phenomenon can be observed clearly in the red digits:  0-4 are more dispersed (consider the 0's, 1's and 3' for the most obvious examples) than the rather cluttered 5-9. The fact that the shared branches map 0-4 and 5-9 much more closely than the private ones is quantified in the paper. Note that this is distinct from the observation that the common branches containing
> the shared layer (green and magenta) are much more mixed between themselves than the private branches (red and black). See also reply to AnonReviewer3. However, we agree that the figure is confusing, and it will be reworked. In particular, we intend to split it into four separate ones, for each branch, as well as add more visual evidence for our beliefs.
>
> * From reducing the capacity of JAE to be the same as the baseline, most of the improvement is gone.
>
> The reduced-capacity JAEs still retain over two thirds (22-24% vs 33-37%) of the observed advantage, therefore most of the advantage remains.
>
> * It is not clear how much of the improvement will remain if the baseline model gets to see all the samples instead of just those from each sub-task
>
> The baseline models, as a pair, see all of the samples the JAE model sees.
>
> * section 4.2.1: The authors demonstrate the influence of shared layer depth in table 2. While it does seem to matter for tasks of dissimilar inputs, have the authors compare having a completely shared branch or sharing more than just a single layer?
>
> We did perform various comparisons between different sharing strategies, but so far could not discern an obviously superior option. However, it remains an intriguing question that we will be paying attention to in future research.
>
> * The authors suggested in section 4.1 CIFAR experiment that the proposed method provides more performance boost when the two tasks are more similar, which seems to be contradicting to the results shown in Figure 3, where its performance is worse when transferring between USPS and MNIST, which are more similar tasks vs between SVHN and MNIST. Do the authors have any insight?
>
> Regarding the surprisingly good performance on the SVHN->MNIST task (vs. the CIFAR experiments), the explanation is the setting. Following established protocol (e.g., [3]), we perform the MNIST<->USPS tasks with small subsets of the datasets, whereas SVHN->MNIST is done using the entire dataset.
>
> See also our reply concerning labeled set size flexibility and transfer learning with multiple tasks - challenges we are able to handle far more naturally than competing approaches.
>
> [4] Weston, Jason, et al. "Deep learning via semi-supervised embedding." Neural Networks: Tricks of the Trade. Springer Berlin Heidelberg, 2012. 639-655.
> [5] S. Parameswaran and K. Q. Weinberger, “Large margin multi-task metric learning,” NIPS 23, pp. 1867–1875, 2010.
> [6] Dumoulin at al., Adversarially Learned Inference, https://arxiv.org/abs/1606.00704
> [7] Devroye, L., Gyoörfi, L., and Lugosi, G. (1996). A Probabilistic Theory of Pattern Recognition. Springer.

---

### Official Review · AnonReviewer2 · 2017-11-28
**The paper proposes a model for allowing various deep neural network architectures to share weights (parameters) across different datasets. The authors then apply the framework to transfer learning.**

**Rating:** 5
**Confidence:** 3

**Review:**

The paper addresses the question of identifying 'shared features' in neural networks trained on different datasets.  Concretely, suppose you have two datasets X1, X2 and you would like to train auto-encoders (with potential augmentation with labeled examples) for the two datasets. One could work on the two separately; here, the authors propose sharing some of the weights to try and exploit/identify common features between the two datasets. The authors formalize by essentially looking to optimize an auto-encoder that take inputs of the form (x1, x2) and employing architectures that allow few nodes to interact with both x1,x2. The authors then try to minimize an appropriate loss function by standard methods.

The authors then apply the above methodology to transfer learning between various datasets. The empirical results here are interesting but not particularly striking; the most salient feature perhaps is that the architectures and training algorithms are perhaps a bit simpler but the overall improvements over existing methods are not too exciting.

---

> ### Author Response · Authors · 2017-12-24
> **Response to AnonReviewer2**
>
> * The empirical results here are interesting but not particularly striking; the most salient feature perhaps is that the architectures and training algorithms are perhaps a bit simpler but the overall improvements over existing methods are not too exciting.
>
> We believe the architectures and training we use are a lot simpler than most comparable methods. For instance, our model for SVHN->MNIST is an order of magnitude smaller than [1], and we do not require a GAN.
>
> See also our reply concerning labeled set size flexibility and transfer learning with multiple tasks - challenges we are able to handle far more naturally than competing approaches.

---

### Official Review · AnonReviewer3 · 2017-12-05
**An appealing architecture for domain adaptation, multitask, and transfer learning but without strong enough results**

**Rating:** 5
**Confidence:** 4

**Review:**



The paper focuses on learning common features from multiple domains data in a unsupervised and supervised learning scheme. Setting this as a general multi task learning, the idea consists in jointly learning autoecnoders, one for each domain, for the multiples domain data in such a way that parts of the parameters of the domain autoencoder are shared. Each domain/task autoencoder  then consists in a shared part and a private part. The authors propose a variant of the model in the case of supervised learning and end up with a general architecture for multi-task, semi-supervised and transfer learning.

The presentation of the paper is good and the paper is easy to follow and explores the rather intuitive and simple idea of sharing parameters between related tasks.

Experimental show some interesting results. First unsupervised experiments on Mnist data show improved MSe of joint autoecnoders but are these differences really significant (e.g. from 0.56 to 5.52) ?  Moreover i am not sure to understand the meaning of separation criterion computed on t-sne of hidden representations. Results of Table 1 show improved reconstruction performance (MSE?) of joint auto encoders over independent ones for unrelated pairs such as airplane and horses. I a not sure ti understand why this improvement occurs even with very different classes.    The investigation on the depth where sharing should occur is quite interesting and related to the usual idea of higher transferable property low level features. Results on transfer are the most interesting ones actually but do not seem to improve so much over baselines.

---

> ### Author Response · Authors · 2017-12-24
> **Response to AnonReviewer3**
>
> * First unsupervised experiments on Mnist data show improved MSe of joint autoecnoders but are these differences really significant (e.g. from 0.56 to 5.52) ?
>
> We agree that MNIST does not show a lot of improvement, due to its simplicity. Note that our experiments with CIFAR-10 display a significant advantage for the JAE scheme.
>
> * Moreover i am not sure to understand the meaning of separation criterion computed on t-sne of hidden representations.
>
> We expect the shared branches to map the inputs to relatively similar hidden states, as they both capture the joint features from both datasets. Following the same logic, the task-specific branches should map inputs to relatively distinctly – they learn different mappings and should not be similar. The statistical measure of this difference is given by the Fisher separation criterion, which is indeed small for the shared branches and large for the private ones.
>
> * Results of Table 1 show improved reconstruction performance (MSE?) of joint auto encoders over independent ones for unrelated pairs such as airplane and horses. I a not sure ti understand why this improvement occurs even with very different classes.
>
> Our explanation for the experienced improvement, even with very different classes, is that the various classes of natural images as captured by the CIFAR-10 dataset share "deep" features necessary for successful reconstruction. We certainly agree that more similar classes should share more of these features, and our results support this intuition.
>
> * Results on transfer are the most interesting ones actually but do not seem to improve so much over baselines.
>
> We agree that some of the improvements over existing methods are modest, though by no means all (e.g., SVHN->MNIST, Fig. 3.c). However, we would like to point out that the methods we compare ourselves to either use large, complicated architectures, require computationally expensive training, or both. We believe that the fact that we out-perform such state-of-the-art approaches with a simple concept while also employing much smaller models is compelling evidence in favor of the shared-subspace hypothesis. Moreover, the ability to perform domain adaptation without training a GAN should be of interest, as most successful state-of-the-art methods require training at least one GAN, a notoriously challenging task.
>
> See also our reply concerning labeled set size flexibility and transfer learning with multiple tasks - challenges we are able to handle far more naturally than competing approaches.

---

### Author Response · Authors · 2017-12-24
**Thank You For The Thoughtful Reviews**

We thank the reviewers for the various points raised. We will reply to each review separately; however, we would like first to point out a contribution of our work that we believe bears stressing. Among the works with similar approach and comparable performance to ours, most seem to be unable to handle more than two tasks (e.g., transfer learning from two sources to a target  ) without either a significant increase in complexity or some novel ideas. [1] would require a number of loss functions growing quadratically in the task number, and an even more demanding architecture  than they already use. [2] would require a quadratically growing amount of discriminators, or else a novel idea to perform efficient domain adaptation for multiple tasks. It is even less clear how to extend [3] to such scenarios.
In contrast, the approach we propose handles this task in stride, simply adding a branch to the joint autoencoder. The experiments in Sec. 4.2.3 support this claim. We believe that this property of joint autoencoders is not matched by any comparable approach, and consider this to be a key advantage of the proposed method.

In addition, we are able to deal with a more flexible range of labeled sample sizes than the aforementioned papers, some of which are not capable of making immediate use of labeled data.

[1]  Bousmalis, K. et al. (2016). Domain separation networks. Advances in Neural Information Processing Systems 29 (NIPS 2016)
[2] Liu, M.-Y. and Tuzel, O. (2016). Coupled generative adversarial networks. In Advances in Neural Information Processing Systems, pages 469–477.
[3] Tzeng, E., Hoffman, J., Saenko, K., and Darrell, T. (2017). Adversarial discriminative domain adaptation. CoRR abs/1702.05464.

---

### Decision · Program_Chairs · 2018-01-29
**ICLR 2018 Conference Acceptance Decision**

**Decision:**

Reject

**Comment:**

Thank you for submitting you paper to ICLR. ICLR. The consensus from the reviewers is that this is not quite ready for publication. In particular, the experimental results are promising, but further work is required to fully demonstrate the efficacy of the approach.